# Recent Insights into Insect Olfactory Receptors and Odorant-Binding Proteins

**DOI:** 10.3390/insects13100926

**Published:** 2022-10-13

**Authors:** Tal Soo Ha, Dean P. Smith

**Affiliations:** 1Department of Biomedical Science, College of Natural Science, Daegu University, Gyeongsan 38453, Gyeongsangbuk-do, Korea; 2Departments of Pharmacology and Neuroscience, University of Texas Southwestern Medical Center, Dallas, TX 75390, USA

**Keywords:** olfaction, odorant-binding proteins, odorant receptors, olfactory neurons, disease vector

## Abstract

**Simple Summary:**

Starting with efforts to understand pheromone detection in moths, the ability of insects to localize food and mates has captivated researchers for over a century. Here we review recent advances in understanding the detection of volatiles at the level of olfactory receptors and odorant-binding proteins focusing on advances in biological understanding. *Drosophila* remains an important model system, but CRIPSR-mediated genome editing is opening the door to genetic analysis in a wide range of insects, including disease vectors and important agricultural pests. This review will spotlight new findings for the initial recognition and signaling events by odorant-binding proteins and olfactory receptors in responses to odorants. Finally, we discussed how some critical insect behaviors appear to have evolved multimodal mechanisms to maximize response robustness.

**Abstract:**

Human and insect olfaction share many general features, but insects differ from mammalian systems in important ways. Mammalian olfactory neurons share the same overlying fluid layer in the nose, and neuronal tuning entirely depends upon receptor specificity. In insects, the olfactory neurons are anatomically segregated into sensilla, and small clusters of olfactory neurons dendrites share extracellular fluid that can be independently regulated in different sensilla. Small extracellular proteins called odorant-binding proteins are differentially secreted into this sensillum lymph fluid where they have been shown to confer sensitivity to specific odorants, and they can also affect the kinetics of the olfactory neuron responses. Insect olfactory receptors are not G-protein-coupled receptors, such as vertebrate olfactory receptors, but are ligand-gated ion channels opened by direct interactions with odorant molecules. Recently, several examples of insect olfactory neurons expressing multiple receptors have been identified, indicating that the mechanisms for neuronal tuning may be broader in insects than mammals. Finally, recent advances in genome editing are finding applications in many species, including agricultural pests and human disease vectors.

## 1. Introduction

Insect chemical detection has intrigued scientists for over 150 years. One of the first descriptions of insect chemotaxis comes from the writings of Jean-Henri Fabre in ’The Life of a Caterpillar’ in 1916 [1]. He described earlier observations in which a captive female Great Peacock Moth in a wire cage attracted 40 male moths of the same species through an open window in his study. He performed the first experiments, repeating the trial with a cloth over the cage, which did not impede the attraction, while a female moth sealed in a tin abolished the attraction. He concluded that there must be an odor emitted from the female moth that was the cause of the male attraction that he was unable to perceive. A half century later, Adolf Butendant succeeded in isolating Bombykol, the first pheromone, from the silkworm moth *Bombyx mori* [2]. The work by Kaissling and Priesner in 1970 revealed that the sensitivity of Bombyx pheromone detection approached perfection, showing that a few molecules of pheromone are sufficient to activate the pheromone-sensing neurons [3]. Insight into the mechanisms for how this remarkable sensitivity is achieved at the molecular level has advanced but is still not entirely understood.

*Drosophila* has served as a potent investigational tool to discover the essential components important for pheromone and odorant detection, to elucidate the computational logic of olfactory neuron innervation, and for exploring the high processing mechanisms by the brain [4]. While spearheaded by *Drosophila*, the advent of CRISPR technology has expanded the range of species amenable to genetic targeting, providing a much more reliable loss-of-function phenotype compared with RNA interference methods. Indeed, genetic lesioning in *Aedes* and *Anopheles* mosquitoes [5,6,7] and moths, including *Bombyx* [8], *Helicoverpa* [9], *Manduca* [10,11,12], and others, contributes to our understanding of these remarkable chemical detectors. This review will provide context for recent findings in the insect chemosensation field.

## 2. Insect Olfactory Receptors (Ors, Irs, and Grs)

Genetic screens in *Drosophila* designed to recover odorant-insensitive mutants identified some genes involved in chemosensation but failed to identify the olfactory receptors [13,14,15]. In hindsight, since typical odorants activate more than one receptor, the mutation of any single receptor probably has a small effect on olfactory behaviors. Furthermore, there are two major classes of olfactory receptors (Ors and Irs), so even mutations in coreceptor subunits were missed.

When candidate olfactory receptors were identified in vertebrates [16], there was great hope in the invertebrate chemosensory community that this would directly lead to the identification of the insect odorant receptors for volatile chemicals. Low stringency screens and degenerate PCR approaches to identify receptors based on G-protein-coupled receptor homology all failed. Success finally came when early drafts of the *Drosophila* genome project became available that were screened using bioinformatics for candidate multiple transmembrane domain proteins. In situ hybridization was used to identify the expression patterns of these candidates, and some were restricted to the antenna [17,18]. At the same time, Richard Axel’s group sequenced cDNAs enriched for rare antenna-specific transcripts from antennal tissues for candidates and identified receptors as well [19]. The *Drosophila* receptors had virtually no homology with the vertebrate counterparts, and receptor topology studies indicated that, unlike GPCRs, the C-terminus of these seven transmembrane receptors is extracellular and thus is reversed in the membrane compared with typical GPCRs [20]. Subsequent studies revealed that the receptors are ligand-gated ion channels [21,22,23]. In *Drosophila* and most higher insects, the Or tuning subunits form heteromers with a common, highly conserved subunit, Or coreceptor (Orco) [24,25]. It is likely that Orco evolved as a coreceptor to allow for a common odorant sensitivity regulation mechanism independent of the tuning receptor expression [26,27,28,29].

The second major family of olfactory receptors found in most insects are the ionotropic receptors (Irs) [30]. These receptors are distinct from the Or family, do not use Orco, and are related to ionotropic glutamate receptors. This family may predate the Or group, as lobsters express Irs but not Ors [31]. The *Drosophila* Ir family has 63 members, and most of the olfactory or antennal Irs primarily detect acids and amines [32,33]. Several members of the Ir family are widely distributed in the olfactory system, specifically Ir8a, Ir25a, and Ir76b, and appear to be Ir coreceptor subunits present in neurons with diverse sensitivities. Finally, a third class of receptors is primarily found in the taste neurons of *Drosophila*, the Grs. However, two Grs members Gr21a and Gr63a mediate CO_2_ sensitivity in an olfactory neuron in the *Drosophila* antenna [34,35]. Thus, three families of receptors mediate the detection of volatiles in *Drosophila*.

## 3. Structural Studies

Recently, cryo-EM structural analysis of two invertebrate olfactory receptors was solved. The Ruta lab studied the homomeric Orco structure of the wasp *Apocryta bakeri* by cryo-EM [36] and, more recently, the homomeric MhOR5 receptor from the jumping bristletail *M. hrabei* [23]. This latter insect lacks Orco, and the expression of MhOR5 alone was capable of conferring GCaMP-mediated odorant responses to eugenol and DEET when expressed in HEK cells. Both investigations established the functional channel as a tetramer. The results of the structure analysis with the MhOR5 receptor with activating ligands revealed that odorants indeed open the pore of the tetrameric structure. These studies set the stage for future work elucidating receptor structures with Orco with tuning receptor heteromers and the structure and subunit composition of functional Irs.

## 4. Olfactory Neuron Tuning

The landmark finding following the identification of vertebrate olfactory receptor genes was the demonstration that a single allele of a single receptor gene is expressed in any single neuron, and all the neurons expressing that allele synapse at one of two glomeruli in the olfactory bulb [37]. The vertebrate olfactory neuron axon targeting the same glomerulus is dependent on the olfactory expression [38], so the expression of multiple receptors in a single cell would be detrimental for odorant discrimination. By contrast, the receptor mutants in *Drosophila* still project their axons to the correct glomerulus, revealing no allelic exclusion and no role for olfactory receptors in pathfinding [39,40,41].

Since insect olfactory neurons do not use olfactory receptors for axonal wiring, this frees these neurons to express multiple receptors. In *Drosophila* olfactory receptor expression mapping studies, most neurons appear to express single receptors. However, at least one neuron expresses three tuning receptor subunits, Or65a, Or65b, and Or65c [42]. These receptor genes are clustered in the genome, so this coexpression could simply represent shared promoter elements from recent gene duplication events. However, these receptors are divergent and likely to respond to different ligands [43]. The olfactory neurons expressing these three receptors have a special role. It was shown that mating silences the DA1 glomerulus (the target for cVA pheromone-sensing Or67d neurons), and Or65 neurons mediate this suppression [44]. Furthermore, courtship behavior is suppressed by the activation of Or65-expressing neurons [43]. These findings suggest that volatiles associated with mating behavior activate Or65 neurons and suppress Or67d activity. Whether all or a subset of the Or65 tuning receptors function to mediate this response is unknown. However, multiple receptors can be coexpressed in the same olfactory neuron. Indeed, Or69aA and Or69aB are coexpressed in the same olfactory sensory neurons and respond to food and putative pheromone odorants, respectively [45]. In mosquitoes, multiple olfactory receptor subunits clustered in the genome are coexpressed and respond to a broad spectrum of odorants [46].

In the past year, receptor coexpression has been expanded to receptors from different classes. By replacing the coding sequence of several widely distributed olfactory receptor coreceptors including Orco, Ir8a, Ir76b, and Ir25a with yeast or neurospora transcription factors and sensitive reporters, the Potter group found an extensive expression overlap among these coreceptors [47]. Indeed, Ir25a is coexpressed with 82% of Orco-expressing neurons. While loss of Orco silenced these neurons, loss of Ir25a often reduced or increased the amplitude of the odorant responses [47]. How the Irs affect the Or and Orco responses is not clear—it seems unlikely that the Irs multimerize with these subunits because changes in odorant responses would be expected to be more dramatic. There may be some complex voltage dependence among these receptors that explains this finding, but future studies will be needed to sort this out.

Recently, the Vosshall lab looked at the receptor expression in mosquitoes. Using single-nucleus RNA sequencing of antenna and palp olfactory neurons, they found coexpression of up to six receptor subunits in single cells, including Ors, Irs, and even Grs [48]. One can envision several scenarios for expressing multiple receptors in single neurons. There could be ligands only detectable with a complex subunit composition, or different subunits could impart a wider dynamic response range to a critical odorant. Indeed, CO_2_ detection in mosquitoes involves three Gr members, Gr1, Gr2, and Gr3 [49]. Whereas Gr2 and Gr3 are sufficient to detect CO_2_, Gr1 expands the dynamic range [49,50].

Expressing more than one receptor could expand the range of activating ligands for the neuron, allowing it to respond to ligands detected by either receptor (an ‘or’ gate) (Figure 1b, top panel). Alternatively, neuronal activation to specific ligands could require activation of both receptors (an ‘and’ gate), only firing in the presence of both ligands (Figure 1b, bottom panel). In principle, an ‘and’ gate would impart high selectivity, whereas an ‘or’ gate would expand the odor response space of the neuron. In the Vosshall study, the ‘or’ gate hypothesis seems to be true. By expressing both Ir and Or receptors in palp neurons that detect skin amines and 1-octen-3-ol, respectively, these neurons respond to either ligand and trigger attraction. Indeed, this illustrates that a robust outcome—identifying a blood source—is more important than discriminating skin components.

## 5. Pheromone Receptors

A number of candidate pheromone receptors were identified in the early 2000s [51,52], and the heterologous expression of the receptors conferred some pheromone sensitivity in tissue culture cells or *Xenopus* oocytes supporting the idea that these are responsible for pheromone sensitivity [53,54,55].

The first in vivo demonstration for the identification of a pheromone receptor was in *Drosophila*. A developmental mutant lacking cVA-sensitive neurons was used to identify pheromone receptors present in wild-type antennas lacking in the mutant by RT-PCR [56]. The Or67d receptor was missing in the mutant antenna, and it was shown that this is the in vivo tuning receptor subunit for cVA detection by demonstrating that it is sufficient to confer cVA pheromone sensitivity to *Drosophila* aT4 trichoid neurons normally insensitive to the pheromone [56]. Interestingly, cVA responses in the aT4 neurons were still dependent on LUSH, a cVA-binding member of the OBP family (see below) [56]. At about the same time, mutants in Or67d were recovered in a single sensillum electrophysiology screen for cVA-insensitive mutants [57]. This was the first insect receptor mutant recovered in a genetic screen. This allele has a point mutation in Or67d that changed a cysteine to tryptophan at amino acid 23 that is expressed but completely insensitive to cVA pheromone.

A second group also identified Or67d as a pheromone receptor. The Dickson group identified Or67d as a receptor expressed in the pheromone-sensitive neurons by RNA in situ hybridization and generated mutants in the gene that are cVA insensitive [40]. These researchers also showed that the mutants had a similar phenotype to *lush* mutants, with reduced copulation rates and increased male–male courtship [40].

In addition to mutants in Or67d, genetic screens recovered a mutation in the *Snmp1* gene that is cVA unresponsive [57]. Evidence suggests that Snmp1 is a component of the pheromone receptor and is not required for the detection of most odorants. Snmp1 was discovered by Richard Vogt’s group by characterizing antisera from mice immunized with moth pheromone-sensing dendrites [58,59]. Snmp1 encodes a CD36 homolog related to scavenger receptors important for cholesterol metabolism [60]. *Drosophila Snmp1* mutants were also reported by the Vosshall group [57,61]. These mutants lack cVA responses and had high constitutive activity in the pheromone-sensitive neurons. This defect could be phenocopied by infusing antibodies against Snmp1 into the sensillum lymph, revealing that Snmp1 is on the dendritic surface [57]. Split GFP experiments indicated that Snmp1 is associated with the pheromone receptor complex [61]. Both studies concluded that Snmp1 acts to deactivate these neurons (Figure 2). Subsequent studies by the Montell group showed that *Snmp1* mutant virgin female insects, never exposed to cVA, had low activity in Or67d neurons and gradually responded to cVA after several minutes of exposure [62]. Remarkably, once the pheromone-sensitive neurons are activated, they continue to fire at high frequency, even long after the removal of the stimulus [62]. Together, these studies suggest that Snmp1 is important for both the activation and deactivation of cVA responses. The current thinking is that Snmp1 is important for loading and unloading the pheromone from the receptors. How Snmp1 fits into the receptor complex at the molecular level is unknown.

Another factor affecting odor and pheromone sensitivity is dATP8B. The phospholipid flippase ATP8B is required in Orco-expressing olfactory neurons for full odor sensitivity [63,64]. This enzyme belongs to the P4-type ATPase family and thought to flip phosphatidylserine from the outer to inner leaflet of the membrane, where it may affect channel proteins in the membrane [65] or affect the localization of a PKC known to phosphorylate Orco for maximum odor sensitivity [29]. How the lipid composition in the dendritic membrane affects olfactory receptor signaling mechanisms requires further study.

Pheromone receptor homologs have now been identified for many species of insects, including pest species and human disease vectors [45,66,67,68,69]. However, putative insect olfactory receptors, including pheromone receptors, are not highly conserved among insect species [70]. A few are proven to be bona fide pheromone receptors by CRISPR-mediated mutation producing a chemosensory deficit (for example [9]).

## 6. Odorant-Binding Proteins (OBPs)

Once moth pheromones were isolated and chemically identified, it became possible to synthesize these molecules. Vogt and Riddiford generated radiolabeled pheromone from the moth *Antheraea polyphemus* with the intent of identifying pheromone receptors in the male antenna [71]. They identified a small, 15 kD soluble, pheromone-binding protein specific to the male antenna that bound the labeled pheromone in native gels [71]. While they did not identify the elusive pheromone receptor, this work was a milestone because it identified the first member of the invertebrate odorant-binding protein family. In *Drosophila*, the number of odorant-binding protein genes rivals that of tuning Or receptors. However, not all are restricted to the olfactory system [72]. The function of odorant-binding proteins remained a puzzle for several decades.

Five hypotheses were proposed for possible roles for these abundant extracellular proteins that directly interact with odorant ligands [73,74]. The first is they are carriers to transport the hydrophobic pheromone molecules through the aqueous sensillum lymph. A second possibility is that they are selectivity filters that concentrate biologically relevant ligands in the lymph. A third possibility is that they present the odorant ligands to the receptors. Fourth, it was postulated that these proteins clear the sensillum lymph of stray molecules, and they could bind and deactivate odorants, by sequestration. Finally, they could function to deliver ligands to degrading enzymes following receptor activation [73,74].

The first clue into the odorant-binding protein function came from Blanka Popoff, who showed that in open-tip preparations, both pheromone and pheromone-binding protein were important for neuronal activation [75]. This suggested that the OBPs were important for neuronal activation, theoretically eliminating odorant degradation and sequestration models. A role for the activation of olfactory neurons was clearly demonstrated when the first *Drosophila* OBP mutant was generated. Mutants lacking the OBP LUSH are insensitive to the concentrations of the male-specific pheromone 11-*cis* vaccenyl acetate (cVA) that potently activate wild-type neurons expressing the Or67d receptor subunit [76]. One of the major behavioral defects associated with *lush* mutants is a reduction in courtship and reduced male–female discrimination in courtship behaviors [77]. When very high concentrations of cVA are applied to the *lush* mutants, weak electrophysiological responses are observed revealing that LUSH is not absolutely required for the activation of pheromone-sensing neurons but is essential for the activation of the olfactory neurons to concentrations of pheromone relevant for single-fly social interactions [78].

A second *lush* mutant phenotype was observed in the single sensillum electrophysiological recordings that provides another valuable clue. In the absence of pheromone, the spontaneous activity in the pheromone-sensitive neurons is reduced 400-fold in *lush* mutants compared with that in wild-type neurons [76]. This unexpected phenotype was not rescued by the expression of a moth pheromone-binding protein, and no effect was observed on *Drosophila* aT4 neurons that do not respond to cVA but still express LUSH. Remarkably, infusing recombinant LUSH protein through a recording pipet is sufficient to restore normal spontaneous activity and, eventually, cVA responses [76]. Therefore, the LUSH protein itself is somehow triggering the activity in pheromone-sensing neurons in the complete absence of pheromone! Together, these data are consistent with a combination of the carrier and presentation models for OBP function and implicate a direct interaction between the LUSH and Or67d–Orco pheromone receptors on the dendrites. Following these studies, many groups have demonstrated specific roles for OBP members in the detection of volatile ligands in many species [79,80,81,82,83,84,85,86,87,88], and even for tastants and humidity [89,90]. Finally, in *Drosophila*, an odorant-binding protein has been implicated in blocking sugar taste detection when sucrose is present with bitter compounds [91]. OBP49a binds bitter ligands and is secreted into the sensillum lymph of sucrose-detecting L-type sensilla in the proboscis. In the presence of sucrose, the neurons robustly respond. However, if bitter compounds are mixed with the sucrose, neuronal activation is blocked, and this modulation requires OBP49a. This suggests a model in which OBP49a, when liganded to bitter molecules, blocks the ability of sucrose to activate the neurons and supports the notion that binding proteins can directly interact with receptors.

Structural studies have revealed that odorant-binding proteins, despite their highly diverse primary sequences, fold into similar globular structures with a central cavity. The structural similarity stems from the disulfide bridges between cysteine residues spaced at conserved intervals (reviewed in [92]). The conserved structure with diverse primary sequence is consistent with a role in direct interactions with structurally diverse odorant molecules. Several groups have solved the structure of odorant-binding proteins with and without bound ligands [93,94,95,96,97,98]. However, odorant-binding proteins tend to be relatively promiscuous when it comes to ligand binding, and simply binding an odorant does not necessarily correlate with the biological function. For example, in addition to cVA pheromone, LUSH binds a variety of molecules including phthalates [99]. However, phthalates do not activate the pheromone-sensitive neurons, whereas cVA does. For pheromones, binding often induces a conformational change [78,97,98]. Attempts to identify interactions with ligand-bound odorant-binding protein conformations have not detected direct interactions with receptors. This could reflect a weak interaction with these membrane receptors, or perhaps the binding proteins simply deliver their cargos to the dendritic membrane, where it could be captured by Snmp1 in the receptor complex. pH-dependent conformational changes have been demonstrated in moth pheromone-binding proteins, leading to the suggestion that pheromone release is triggered by proximity to charged phospholipids [97,100]. Whatever model is proposed for odorant-binding proteins and pheromone sensitivity, it should be able to account for the striking loss of spontaneous activity in *lush* mutant pheromone-sensing neurons in the absence of pheromone [76].

## 7. Recent Advances in Odorant-Binding Proteins

In addition to LUSH, the trichoid sensilla in *Drosophila* also secrete at least two additional odorant-binding proteins, Os-E and Os-F [101]. While LUSH expression is restricted to the trichoid sensilla, Os-E and Os-F are expressed in both trichoid and intermediate sensilla. Os-E and Os-F are in close proximity in the genome, allowing the use of CRISPR to delete both binding protein genes [102]. These mutants have increased electrophysiological responses to farnesol, a ligand abundant in citrus peels and detected by neurons expressing Or83c receptors [103]. However, from the results of the kinetic analysis, it was obvious that the increased action potentials per second are actually due to the defective deactivation of these neurons in response to the odorant. Single sensillum recordings reveal prolonged activity in the firing of these neurons in *Os-E,Os-F* double mutants compared with controls [102]. Farnesol and 3-hexanol both activate Or83c receptors. Remarkably, while responses to farnesol are prolonged in the Os-E/Os-F double mutant, the responses to 3-hexanol are unaffected by the loss of these OBPs. This reveals that there are odorant-specific defects in the OBP mutant even for odors that activate the same receptor. Os-E and Os-F are involved in the deactivation of farnesol responses perhaps by unloading the Or83c receptors of this hydrophobic ligand. This mechanism is quite different from that observed with *lush* mutants and is more consistent with an odorant removal mechanism. The Carlson lab reported that loss of an abundant odorant-binding protein restricted to ab8 sensilla, OBP28a, does not affect the amplitude of odorant responses of the ab8 neurons but may act as a buffer for odorants [104].

Finally, an intriguing trend is becoming evident: that is, some odorant-binding proteins are associated with insecticide resistance. In the Asian citrus psyllid, *Diaphorina citri*, OBP2 is induced by exposure to imidacloprid, and knockdown of this OBP by RNA interference increased susceptibility to this insecticide [105]. OBP2 binds well to imidacloprid and at least two other neonicotinide insecticides [105]. In *Culex quinquefasciatus* mosquitoes, OBP28 is protective against deltamethrin, and knockdown of this OBP greatly enhances susceptibility to this insecticide [106]. OBPc13 is induced by *Artemisia vulgaris* oil in the stored grain pest *Tribolium castaneum*, and RNA interference against this OBP increases susceptibility to the oil [107]. In the rice pest *Nilaparvata lugens*, OBP3 is associated with nitenpyram and sulfoxaflor resistance [83]. It is likely that these OBPs are sequestering the insecticide, preventing the compounds from access to their site of action. If true, this would reveal yet another mechanism of action for OBPs. Nature appears to have taken full advantage of possible OBP functional roles.

## 8. CRISPR (Clustered Regularly Interspaced Short Palindromic Repeats)

Pheromone receptors expressed in heterologous systems such as Sf9 cells, HEK293 cells, or *Xenopus* oocytes do not always show the same specificity as they do in vivo. For example, *Epiphyas postvittana* Or1 expressed in Sf9 cells responds to plant volatiles but not to pheromones, despite mapping to the pheromone receptor clade [108]. However, when expressed in HEK293 cells, it responds to pheromone but not plant volatiles [109,110]. Pheromone receptors from *Eriocrania semipurpurella* distinctly behave when expressed in HEK293 cells and *Xenopus* oocytes [67,111]. Therefore, it has become clear that in vivo demonstration of function is critical to assign specific gene products to chemical detection in intact animals.

RNA interference, produced by injecting double-stranded RNA homologous to specific genes into animals, has provided some reduced function, but knockdown is not complete. For many factors, this may simply not be sufficient to observe a phenotype. The combination of available genome sequences for many insects and the availability of the CRISPR–Cas9 system is revolutionizing the way gene products are studied in agricultural pests and human disease vectors. This system is amenable to any insect system, making virtually any species a genetic model system. CRISPR allows lesions to be generated in specific genes by introducing the Cas9 nuclease into embryos and targeting the nuclease to specific genes with a 20 nucleotide RNA homologous to the target adjacent to a PAM site (NGG) [112]. By introducing two cleavage sites in the gene, the target genes can even be replaced with an RFP expression cassette, allowing ease of identifying mutants [113]. We expect this approach to become the gold standard for demonstrating the functional relevance of specific gene products in chemical sensation. Indeed, CRISPR-mediated mutants in Orco have already been reported for several insect pests [6,114,115,116,117].

Finally, the recent work may simplify the generation of CRISPR lesioned animals in different insects by bypassing the need to inject embryos. DIPA-CRISPR involves injecting Cas9 and targeting RNA into the hemolymph of adult female insects [118]. This appears to be sufficient to generate lesions in oocytes and bypasses the requirement to inject embryos. This approach will greatly simplify genetic lesioning in pest species in the future.

## 9. Conclusions

High-throughput DNA sequencing is leading to the identification of olfactory receptors and odorant-binding proteins in diverse insect species. RNA interference approaches that reduce expression provided some in vivo support for function. However, CRISPR gene targeting is become more widespread and is generating complete loss of receptor phenotypes in vivo, and is rapidly replacing RNA interference. DIPA-CRISPR, where adult female insects are injected instead of embryos, will make producing gene-specific lesion even easier in pest species. We can expect rapid progress in new model systems in the next few years.

## Figures and Tables

**Figure 1 insects-13-00926-f001:**
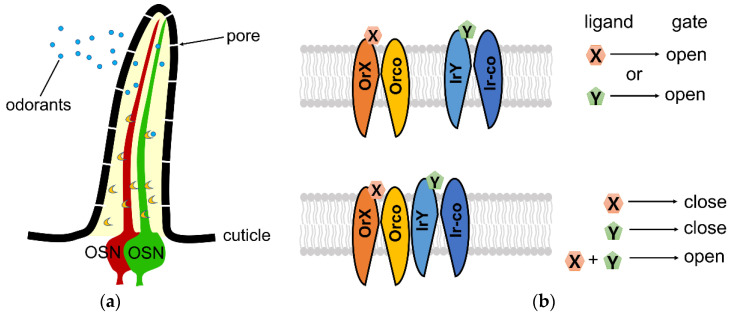
Olfactory receptor organization in insects and tuning models. (**a**) Olfactory organs of Drosophila are covered by sensilla. Each sensillum contains one to four olfactory neurons (OSNs); neurons are shown in red and green. (**b**) Expressing more than one receptor in one neuron leads to ‘or’ gate (top panel) or ‘and’ gate (bottom panel) potential model.

**Figure 2 insects-13-00926-f002:**
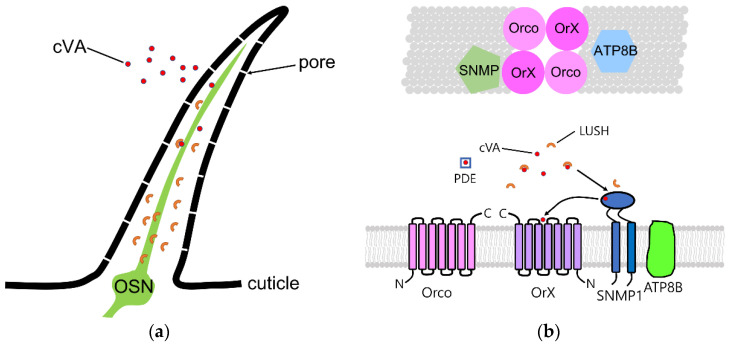
Model of cVA detection mechanism in aT1 neurons. (**a**) Cartoon of aT1 trichoid sensillum. aT1 sensilla have numerous pores in the cuticle layer and house a single olfactory receptor neuron (OSN) dendrite. LUSH (cup-shaped structures in sensillum lymph) binds cVA and deliver it to neuronal receptors. (**b**) Receptor complex in aT1 sensillum. In this sensillum, tetrameric receptor complex is composed of Or67d pheromone receptor, the coreceptor Orco. SNMP1, a CD36-related sensory neuron membrane protein, is also required in dendritic membrane. ATP8B, a lipid flippase, is also important for normal Orco-dependent responses [63,64].

## Data Availability

Not applicable.

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
