# Peer review of "Recent Insights into Insect Olfactory Receptors and Odorant-Binding Proteins"

_insects, 2022, doi:10.3390/insects13100926_

Round 1

Reviewer 1 Report

The authors present a review on recent insights into insect odorant receptors and odorant biding proteins, largely focused on results from the genetic model, Drosophila melanogaster, while expanding on recent efforts utilizing CRISPR Cas9 genome editing to expand olfactory gene studies to non-model systems. The review, generally well written, though it is limited in scope and has the feel more of a mini-review, rather than a comprehensive literature review. Substantial blocks of text are focused on the author’s own previous works (highlighted with terms we/our), including studies that are not so recent. While the authors have sought to aim their focus on recent insights, several recent relevant literature references are missing. Finally, some incorrect or misleading statements are made that will need to be corrected. Revisions are thus required before this manuscript would be suitable for publication.

Line 58. “1” – numericals less than ten should be written out in word form, e.g. “one” in this case.

Line 70. “was performing whole transcriptome sequencing”. This is not the approach that was taken in reference 11. The approach involved differential screening of a cDNA library followed by targeted sequencing of selected clones.

Line 76-77. “the Or tuning subunits form heteromers with a common, highly conserved subunit, Orco”. Critical references are missing for this statement, such as Larsson et al., 2004, Neuron, and Jones et al., 2011, PNAS.

Line 83-84. “The Drosophila Ir family has 63 members that appear to primarily detect acids and amines.” This is not wholly correct. IRs have been revealed to have much more diverse function than just olfaction and detecting amines and acids. Many IRs have been shown to have gustatory function, and several others have functional roles other than chemosensation. Perhaps it was meant to say that most of the olfactory or antennal IRs primarily detect amines and acids…..however it should be recognized that the subset of antennal IRs is a minority of the larger IR family.

In section 4. Olfactory Neuron Tuning, a paragraph is dedicated to phenomena of neurons expressing more than one tuning receptor subunit, references 32-33, which is fine, however, if this review is covering recent insights, this section warrants some mention of the role for alternative splicing of odorant receptor genes and/or expression of transcript variants/isoforms. Evidence for this has been reported on recently in Drosophila melanogaster, Lebreton et al., 2017, BMC Biology.

Furthermore, co-expression of multiple tuning receptor subunits, may also be generalized beyond Drosophila, as reported on in mosquito by Karner et al., 2015, Frontiers in Ecology and Evolution.

Line 166. “five theories”. Should be “five hypotheses”

Furthermore, it is unclear the purpose of Section Five, Odorant Binding Proteins – Early Work. If the theme of this review is on Recent Insights into olfactory receptors and odorant binding proteins, it would seem off topic to dedicate an entire section to “early work”. Instead the early work could be summarized much more briefly as lead in to the “recent advances” section that follows.

In section 6., “recent advances in Odorant Binding Proteins”, a critical reference and discussion therein advancing our knowledge on OBP function in olfaction is missing, namely Larter et al., 2016. If the five hypotheses about OBP function are mentioned (as they should), this report provides sound evidence for OBP function beyond the commonly mentioned role of “transport of OBPs to the membrane”.

Line 255-256. “These workers”. Is it meant to say “these researchers”?

In section 7, most of the discussion on Pheromone Receptors is focused on receptors of Drosophila melanogaster, and for the most part summarizes research that is not representative of recent advances. This text ignores more recent studies of novel pheromone receptors, as in Lebreton et al., 2017, and gives no mention on broadening study of pheromone receptors beyond Diptera, as in Lepidoptera and Coleoptera. See Zhang and Lofstedt, 2015, Frontiers in Ecology and Evolution; Yuvaraj et al., 2017, Molecular Biology and Evolution; Bastin-Heline et al., 2019, eLife, Yuvaraj et al., 2021, BMC Biology. It is critical to mention this, because it is well known that insect odorant receptors, including pheromone receptors, are not generally conserved across insect orders (see Hansson and Stensmyr, 2011, Neuron – Figure 3)

Line 284-285: “However, when expressed in HEK293 cells, it responds to pheromone but not plant volatiles.” A more recent reference for this claim should be cited for EposOR1, namely Yuvaraj et al., 2022, Insect Biochemistry and Molecular Biology.

Line 290-291. “ but generally only reduces gene expression by half.” A statement like this would need to be referenced, especially because knockdown induced by RNAi can often be much greater than half

Author Response

The authors present a review on recent insights into insect odorant receptors and odorant biding proteins, largely focused on results from the genetic model, Drosophila melanogaster, while expanding on recent efforts utilizing CRISPR Cas9 genome editing to expand olfactory gene studies to non-model systems. The review, generally well written, though it is limited in scope and has the feel more of a mini-review, rather than a comprehensive literature review. Substantial blocks of text are focused on the author’s own previous works (highlighted with terms we/our), including studies that are not so recent. While the authors have sought to aim their focus on recent insights, several recent relevant literature references are missing. Finally, some incorrect or misleading statements are made that will need to be corrected. Revisions are thus required before this manuscript would be suitable for publication.

Line 58. “1” – numericals less than ten should be written out in word form, e.g. “one” in this case.

Done.

Line 70. “was performing whole transcriptome sequencing”. This is not the approach that was taken in reference 11. The approach involved differential screening of a cDNA library followed by targeted sequencing of selected clones.

We have corrected this.

Line 76-77. “the Or tuning subunits form heteromers with a common, highly conserved subunit, Orco”. Critical references are missing for this statement, such as Larsson et al., 2004, Neuron, and Jones et al., 2011, PNAS.

Refs added.

Line 83-84. “The Drosophila Ir family has 63 members that appear to primarily detect acids and amines.” This is not wholly correct. IRs have been revealed to have much more diverse function than just olfaction and detecting amines and acids. Many IRs have been shown to have gustatory function, and several others have functional roles other than chemosensation. Perhaps it was meant to say that most of the olfactory or antennal IRs primarily detect amines and acids…..however it should be recognized that the subset of antennal IRs is a minority of the larger IR family.

We have clarified the text.

In section 4. Olfactory Neuron Tuning, a paragraph is dedicated to phenomena of neurons expressing more than one tuning receptor subunit, references 32-33, which is fine, however, if this review is covering recent insights, this section warrants some mention of the role for alternative splicing of odorant receptor genes and/or expression of transcript variants/isoforms. Evidence for this has been reported on recently in Drosophila melanogaster, Lebreton et al., 2017, BMC Biology.

Ref added and text modified

Furthermore, co-expression of multiple tuning receptor subunits, may also be generalized beyond Drosophila, as reported on in mosquito by Karner et al., 2015, Frontiers in Ecology and Evolution.

Ref added.

Line 166. “five theories”. Should be “five hypotheses”

Fixed

Furthermore, it is unclear the purpose of Section Five, Odorant Binding Proteins – Early Work. If the theme of this review is on Recent Insights into olfactory receptors and odorant binding proteins, it would seem off topic to dedicate an entire section to “early work”. Instead the early work could be summarized much more briefly as lead in to the “recent advances” section that follows.

We removed the subtitle. We think the historical context is of interest to the general reader.

In section 6., “recent advances in Odorant Binding Proteins”, a critical reference and discussion therein advancing our knowledge on OBP function in olfaction is missing, namely Larter et al., 2016. If the five hypotheses about OBP function are mentioned (as they should), this report provides sound evidence for OBP function beyond the commonly mentioned role of “transport of OBPs to the membrane”.

Ref added

Line 255-256. “These workers”. Is it meant to say “these researchers”?

changed

In section 7, most of the discussion on Pheromone Receptors is focused on receptors of Drosophila melanogaster, and for the most part summarizes research that is not representative of recent advances. This text ignores more recent studies of novel pheromone receptors, as in Lebreton et al., 2017, and gives no mention on broadening study of pheromone receptors beyond Diptera, as in Lepidoptera and Coleoptera. See Zhang and Lofstedt, 2015, Frontiers in Ecology and Evolution; Yuvaraj et al., 2017, Molecular Biology and Evolution; Bastin-Heline et al., 2019, eLife, Yuvaraj et al., 2021, BMC Biology. It is critical to mention this, because it is well known that insect odorant receptors, including pheromone receptors, are not generally conserved across insect orders (see Hansson and Stensmyr, 2011, Neuron – Figure 3)

Ref added, brief discussed and added to text

Line 284-285: “However, when expressed in HEK293 cells, it responds to pheromone but not plant volatiles.” A more recent reference for this claim should be cited for EposOR1, namely Yuvaraj et al., 2022, Insect Biochemistry and Molecular Biology.

Ref added.

Line 290-291. “ but generally only reduces gene expression by half.” A statement like this would need to be referenced, especially because knockdown induced by RNAi can often be much greater than half

Altered text to read ‘but knock down is not complete’

Reviewer 2 Report

The research progress of olfactory mechanism of insect is very fast in recent years. Therefore, it is of significance to summarize the latest research progress in related fields. In this review, the author summarized and introduced the research progress in two important groups of olfactory genes, OR and OBP. Most of the contents described in this review are accurate, but there are still some issues should be addressed. The detailed comments are as follows:

1. Most of the introductions in this article are about fruit flies, and some about mosquitoes, but few about other insects. This is inconsistent with the title of this review "Recent Insights into Insect Odorant Receptors and Odorant Binding Proteins". In recent years, thanks to the development and application of genomics technology and gene editing technology, the research on olfaction of non-model insects other than drosophila and mosquitoes has made rapid progress. The authors should make necessary introduction in these insects, rather than just focus on model insects

2. The author's using of the two words "odorant receptor " and " olfactory receptor" is somewhat unusual. Since the identification of odorant receptor in drosophila, there is no strict rule on the use of the words " odorant receptor " and " olfactory receptor ". We can even say that the usage of these two words is confused for a long time. But at present, people have formed a basically consistent usage. As described in the review (2018-Fleischer-CMLS-Access to the odor world-olfactory receptors and their role for signal transduction in insects), the olfactory receptor is used as the the general name of receptors involved in olfactory perception, including odorant receptor (OR), IR and GR. This classification is now widely accepted. It’s better that the author can revise the usage of these two words in this article.

3. The order of the subtitles needs to be adjusted. Pheromone receptor belongs to OR. So it is recommended to move the "7. Pheromone receptor " to the front of “5. Odorant Binding Proteins (OBPs)”. Please delete the subtitle "Early work" in line 155.

4. The author's introduction of CRISPR is mainly about the technology, but the successful application of CRISPR in insect (especially non-model insect) olfactory research is not introduced. In fact, the application of CRISPR in non-model insects such as Manduca sexta, Plutella xylostella and Helicoverpa armigera has made a lot of progress in recent years, which should be introduced in this review.

Author Response

The research progress of olfactory mechanism of insect is very fast in recent years. Therefore, it is of significance to summarize the latest research progress in related fields. In this review, the author summarized and introduced the research progress in two important groups of olfactory genes, OR and OBP. Most of the contents described in this review are accurate, but there are still some issues should be addressed. The detailed comments are as follows:

  1. Most of the introductions in this article are about fruit flies, and some about mosquitoes, but few about other insects. This is inconsistent with the title of this review "Recent Insights into Insect Odorant Receptors and Odorant Binding Proteins". In recent years, thanks to the development and application of genomics technology and gene editing technology, the research on olfaction of non-model insects other than drosophila and mosquitoes has made rapid progress. The authors should make necessary introduction in these insects, rather than just focus on model insects.

We added references at the beginning to introduce a wider range of insects.

  1. The author's using of the two words "odorant receptor " and " olfactory receptor" is somewhat unusual. Since the identification of odorant receptor in drosophila, there is no strict rule on the use of the words " odorant receptor " and " olfactory receptor ". We can even say that the usage of these two words is confused for a long time. But at present, people have formed a basically consistent usage. As described in the review (2018-Fleischer-CMLS-Access to the odor world-olfactory receptors and their role for signal transduction in insects), the olfactory receptor is used as the the general name of receptors involved in olfactory perception, including odorant receptor (OR), IR and GR. This classification is now widely accepted. It’s better that the author can revise the usage of these two words in this article.

These terms seem interchangable, and are both commonly used in the literature, but we have changed all 'odorant receptor' terms with 'olfactory receptor'.

  1. The order of the subtitles needs to be adjusted. Pheromone receptor belongs to OR. So it is recommended to move the "7. Pheromone receptor " to the front of “5. Odorant Binding Proteins (OBPs)”. Please delete the subtitle "Early work" in line 155.

Done.

  1. The author's introduction of CRISPR is mainly about the technology, but the successful application of CRISPR in insect (especially non-model insect) olfactory research is not introduced. In fact, the application of CRISPR in non-model insects such as Manduca sextaPlutella xylostellaand Helicoverpa armigerahas made a lot of progress in recent years, which should be introduced in this review.

We have added references and a short paragraph to address this comment.

Reviewer 3 Report

Wouldn't it be more logical to have Section 7 "Pheromone receptors" preceding Section 5 "Odorant Binding Proteins (OBPs)"?

It would be useful to add that some OBPs are expressed in non-olfactory organs as well. See: 

  1. Mastrobuoni G, Qiao H, Iovinella I, Sagona S, Niccolini A, Boscaro F, Caputo B, Orejuela MR, Torre Ad, Kempa S et al608 Proteomic Investigation of Soluble Olfactory Proteins in Anopheles gambiaePLOS ONE 2013, 8(11):e75162

  2. Rihani K, Ferveur JF, Briand L: The 40-Year Mystery of Insect Odorant-Binding ProteinsBiomolecules 2021, 11(4)
  3. Sun JS, Xiao S, Carlson JR: The diverse small proteins called odorant-binding proteinsOpen Biol 2018, 8(12):180208

A brief description of the "classical" OBP fold with appropriate references would be useful e.g. 

  1. Brito NF, Moreira MF, Melo ACA: A look inside odorant-binding proteins in insect chemoreceptionJournal of Insect Physiology 615 2016, 95:51-65

  2. Venthur H, Zhou JJ: Odorant Receptors and Odorant-Binding Proteins as Insect Pest Control Targets: A Comparative 617 AnalysisFront Physiol 2018, 9:1163

Author Response

Wouldn't it be more logical to have Section 7 "Pheromone receptors" preceding Section 5 "Odorant Binding Proteins (OBPs)"?

It would be useful to add that some OBPs are expressed in non-olfactory organs as well. See: 

  1. Mastrobuoni G, Qiao H, Iovinella I, Sagona S, Niccolini A, Boscaro F, Caputo B, Orejuela MR, Torre Ad, Kempa S et al608 Proteomic Investigation of Soluble Olfactory Proteins in Anopheles gambiaePLOS ONE 2013, 8(11):e75162
  2. Rihani K, Ferveur JF, Briand L: The 40-Year Mystery of Insect Odorant-Binding ProteinsBiomolecules 2021, 11(4)
  3. Sun JS, Xiao S, Carlson JR: The diverse small proteins called odorant-binding proteinsOpen Biol 2018, 8(12):180208

Done

A brief description of the "classical" OBP fold with appropriate references would be useful e.g. 

  1. Brito NF, Moreira MF, Melo ACA: A look inside odorant-binding proteins in insect chemoreceptionJournal of Insect Physiology 615 2016, 95:51-65
  2. Venthur H, Zhou JJ: Odorant Receptors and Odorant-Binding Proteins as Insect Pest Control Targets: A Comparative 617 AnalysisFront Physiol 2018, 9:1163

done

Round 2

Reviewer 1 Report

All previous concerns have been addressed in a satisfactory manner. The addition of the figures to the review are a solid improvement to the manuscript. However, one minor correction is needed for the mention of ATP8b and associated references in Figure 2. In this figure, and in the legend, the role of the ATP8b flippase is presented for the first time within the manuscript, though in the figure it is not clear how the flippase is interacting with other components. Furthermore, there is no mention of the flippase at all within the text of the manuscript outside of this figure. A statement about these enzymes (and their role associated with odorant receptors) in the body of the text would be appropriate, if it is worth to mention/show in the figure/legend. Furthermore, the references 57 and 62 have been added for the statement about ATP8b, however, in ref. 57 there is no mention of ATP8b, and in ref. 62, ATP8b is mentioned, but this is not the correct reference for the research done on the flippase. Ref. 62 cites two other articles in the discussion of the flippase; these two articles should be cited here. 

Author Response

All previous concerns have been addressed in a satisfactory manner. The addition of the figures to the review are a solid improvement to the manuscript. However, one minor correction is needed for the mention of ATP8b and associated references in Figure 2. In this figure, and in the legend, the role of the ATP8b flippase is presented for the first time within the manuscript, though in the figure it is not clear how the flippase is interacting with other components. Furthermore, there is no mention of the flippase at all within the text of the manuscript outside of this figure. A statement about these enzymes (and their role associated with odorant receptors) in the body of the text would be appropriate, if it is worth to mention/show in the figure/legend. Furthermore, the references 57 and 62 have been added for the statement about ATP8b, however, in ref. 57 there is no mention of ATP8b, and in ref. 62, ATP8b is mentioned, but this is not the correct reference for the research done on the flippase. Ref. 62 cites two other articles in the discussion of the flippase; these two articles should be cited here. 

Added Rfs and briefly discussed in text. Corrected mis-cited refs in fig 2.